# Marine biogeographic realms and species endemicity

Mark J. Costello [1,2], Peter Tsai[2], Pui Shan Wong[2], Alan Kwok Lun Cheung[3], Zeenatul Basher[1,4] & Chhaya Chaudhary[1]

Marine biogeographic realms have been inferred from small groups of species in particular environments (e.g., coastal, pelagic), without a global map of realms based on statistical analysis of species across all higher taxa. Here we analyze the distribution of 65,000 species of marine animals and plants, and distinguish 30 distinct marine realms, a similar proportion per area as found for land. On average, 42% of species are unique to the realms. We reveal 18 continental-shelf and 12 offshore deep-sea realms, reflecting the wider ranges of species in the pelagic and deep-sea compared to coastal areas. The most widespread species are pelagic microscopic plankton and megafauna. Analysis of pelagic species recognizes five realms within which other realms are nested. These maps integrate the biogeography of coastal and deep-sea, pelagic and benthic environments, and show how land-barriers, salinity, depth, and environmental heterogeneity relate to the evolution of biota. The realms have applications for marine reserves, biodiversity assessments, and as an evolution relevant context for climate change studies.

[1] Institute of Marine Science, University of Auckland, Auckland 1142, New Zealand. [2] Bioinformatics Institute, University of Auckland, Auckland 1142, New Zealand. [3] School of Environment, University of Auckland, Auckland 1142, New Zealand. [4] Center for Systems Integration and Sustainability, Michigan State University, East Lansing, MI 48823, USA. Correspondence and requests for materials should be addressed to M.J.C. (email: m.costello@auckland.ac.nz)

While the occurrences of marine fauna and flora clearly differ between parts of the oceans, whether biogeographic boundaries, and thus definable realms of endemicity, exist has not been clear. Consequently, a centuries old tradition of mapping global marine regions has not produced a single robust regionalization based on empirical species distribution evidence[1, 2]. Indeed, Ekman[1] and Briggs[2] stated that there was little evidence for biogeographic boundaries in the ocean. In contrast, boundaries of terrestrial realms were proposed by Wallace 140 years ago, and recently supported by empirical data analysis[3–7]. If marine boundaries exist, they would indicate the relative importance of factors that have caused the present distribution of marine species at a global scale, such as continental drift, temperature, sea-level rise, and glaciation. Knowledge of the relative endemicity and cosmopolitanism of different taxa, varying in body size, and pelagic and benthic lifestyles, will inform estimates of global species richness because more widespread taxa may be expected to have less species due to higher gene flow[8–10].

Biogeography has been rich in studies of small groups of better known species at local to regional scales[11], with relatively fewer examples of more generalized studies (i.e., across many species and broad spatial scales) and models[3, 12]. An advantage of general models is that they provide a hypothesis that can be falsified, whereas more limited data may not be easily generalized. Indeed, we should expect different groups of species to have different distributions reflecting their evolutionary origins and environmental adaptations. New information on a group of species often complicates previously observed patterns, suggesting that local environmental conditions, including habitat suitability, may have been more important in determining the limits of a species distribution than evolutionary history and climate, and/or that these boundaries were artefacts of the limited data[1, 2, 11, 13]. Because prior grouping of the data before analysis can bias the results[14], caution is necessary while comparing between species groups or pre-defined geographic areas. Rather than using selected taxonomic groups as surrogates for wider biodiversity, when different taxa can show different biogeographies[15], it would be less biased to use all species regardless of their taxonomic classification. To date, global biogeographic reviews have not integrated data across all taxa. Holt et al.[5] classified 11 terrestrial realms (excluding Antarctica) based on the distribution of 21,000 amphibian, bird, and mammal species, which represent < 2% of all terrestrial species. Including invertebrates may further refine and/or subdivide these realms.

Individuals of many marine species drift, swim, or fly across and/or between oceans during their lifetime. These pelagic species, and life stages of many benthic species, contrast with entirely benthic species that spend most of their life on the seabed, and thus may be expected to disperse shorter distances. They also contrast with aerial plankton that is composed of dispersing microbes, plant seeds, invertebrates and their predators (e.g. birds, bats)[16], and perhaps some marine microbiota. While it may be predicted that pelagic species have larger geographic ranges than benthic taxa, whether there is any congruence between pelagic and benthic biogeography is unknown. This led recent reviews to consider pelagic and benthic biogeography separately[17, 18].

The classification of the world into biogeographic realms is of practical interest to many governmental and intergovernmental organisations who wish to identify naturally similar areas for reporting on the state of the environment, for prioritizing conservation action, or providing funding for conservation or economic development[13, 17–22]. Most existing geographic classifications that are in use (e.g., fisheries areas, Longhurst provinces, and Large Marine Ecosystems) are not based on

biogeography and are unlikely to accurately represent the distribution of species and wider biodiversity[21]. Realms contrast with geographic areas defined by communities characterized by their dominant species (i.e. habitats), environment (i.e. ecosystems), and life forms (biomes). These concepts do not consider the endemicity or cosmopolitanism of species[21]. On land such realms are more distinct because the oceans form dispersal barriers that lead to the species evolving in isolation and consequently high endemicity[13].

The absence of suitable global scale maps of marine biogeographic realms led to meetings that proposed and mapped separate regions for the coastal benthos, deep-sea benthos, and pelagos, based on expert opinion[13, 17, 18]. However, these reviews did not conduct a standardized data analysis and a map covering all these environments was not synthesized. In this paper we have integrated data across all these environments to map species endemicity, and show how global patterns of species richness and endemicity compare between coastal and deep-sea, and pelagic and benthic, environments.

Ekman's book[1] was a benchmark in marine biogeography and reviewed about 600 publications up to 1950. It discussed patterns of endemicity at family, genus, and species level for selected taxa. Since 1950 the number of known marine species has doubled[23, 24], and although significantly more distribution data has been collected, it has been scattered in thousands of publications or not published. The recent integration of data sets into standardized databases provides unprecedented access to data across all taxa (e.g., in ref. [23].). The present study provides the first holistic analysis of the Ocean Biogeographic Information System (OBIS)[24, 25], a marine subset of the Global Biodiversity Information Facility that has similarly not yet been analyzed in its entirety, perhaps because the large amount of data (500 million species records in GBIF) is computationally challenging.

In this study, we analyzed the distribution of 65,000 marine species, far more than that in the previous studies, to provide an empirical basis for biogeographic realms. Our analysis is objective in being data driven and reproducible and holistic in covering all accessible data for all taxa in all oceans. We compared the resulting realms to previous biogeographic classifications and propose a new map of marine biogeography that covers all oceans from coastal to the deep sea. This shows how pelagic and benthic biogeographies can be integrated.

## Results

**Seas and oceans.** The first cluster of seas and oceans split the seas at 1% similarity coefficient level into Atlantic-Polar, Black Sea and inner Baltic Sea, and Indo-Pacific realms. These were then subdivided into 10 subrealms (Fig. 1) at higher levels of similarity. That there were no more than three seas in any of the groups of seas shown to be significantly similar by the SIMPROF test (Supplementary Fig. 1) indicated that most areas were different from each other in species composition, and thus biogeographically. The seas that were significantly similar in species composition were all neighbors. Analysis of similarity (ANOSIM) between groups of nearby seas found highly significant differences (R statistic 0.561, and no pairs of groups approached this value, $P < 0.01$), thus re-affirming that the groups represented biogeographically distinct realms.

Clustering the data at genus level was explored to see if higher taxonomic unit revealed the same pattern. The overall structure of the dendrogram was the same as it was for species. Of the 30 realms at species level, over half (17) were the same at genus level, 9 realms had seas added to their group (i.e. genus level were broader), and 2 new groups were formed (Supplementary Table 1). However, seven species-level groups excluded seas from

their group at genus level and three realms were not re-recognized at the genus level. Nearby seas were not always grouped close together, reflecting the low level of similarity between the higher level groups of seas, and thus the sensitivity of the genus-level analysis to small changes in taxonomic composition.

**5° cells**. The biogeographic realms identified by the 5° cells were supported by the groups of seas (Fig. 1) and added additional realms, especially in the open ocean, and including coastal areas of west and southern Africa, southern South America, and New Zealand (Fig. 2). At the 1% level, seven biogeographic realms were distinguished: the freshwater influenced (1) inner Baltic and (2) Black Seas; (3) Arctic-temperate including the North Pacific, North Atlantic, and Mediterranean; (4) mid-tropical North Pacific; (5) south-east Pacific; (6) mid-Atlantic, Pacific, and Indian oceans; (7) Tropical west Pacific coast; and (8) Southern Ocean (Fig. 1). The same analysis for pelagic-only species indicated only five biogeographic realms, comprised of (1) & (2) of the above together, and distinguishing (3), (4), (6), and (8). Analysis of the full data set further subdivided the realms to distinguish 30 biogeographic realms (Fig. 1).

| | | | Seas' group | Realm | % spp unique | # spp |
|---|---|---|---|---|---|---|
| Inner Baltic Sea | | | Inner Baltic Sea | 1 | 63 | 458 |
| Black Sea | | | Black Sea | 2 | 84 | 192 |
| NE and NW Atlantic and Mediterranean, Arctic and North Pacific | NE Atlantic & Mediterranean (2) | NE Atlantic (3) | NE Atlantic | 3 | 27 | 7117 |
| | | Arctic Europe (5) | Norwegian Sea (in part) | 4 | 43 | 1345 |
| | | Mediterranean (3) | Mediterranean | 5 | 45 | 3096 |
| | Arctic & N Pacific (2) | Arctic (3) | Arctic seas | 6 | 19 | 1907 |
| | | North Pacific (3) | N Pacific | 7 | 27 | 5535 |
| | N Atlantic boreal & sub-Arctic from Canada to Greenland Sea (2) | | N American Boreal | 8 | 31 | 1492 |
| Mid-tropical North Pacific Ocean | | | -- | 9 | 47 | 2859 |
| South-east Pacific | | | -- | 10 | 59 | 1618 |
| Mid-Atlantic, Pacific and Indian Oceans including coastal tropics and warm-temperate areas | Tropical W Atlantic & Tropical E Pacific (2) | Tropical W Atlantic (3) | Caribbean & Gulf of Mexico | 11 | 30 | 13281 |
| | | Tropical E Pacific (3) | Gulf of California | 12 | 30 | 3279 |
| | Coastal Indian Ocean, W Pacific, Arabian Gulf to New Caledonia, S Pacific tropical islands, & N, W & E Australia (2) | Tropical Indo-Pacific (East Indies) & coastal Indian Ocean (3) | Indo-Pacific seas & Indian Ocean | 13 | 31 | 16508 |
| | | Red Sea (4) | Gulfs of Aqaba, Aden, Suez, Red Sea | 14 | 74 | 997 |
| | | Tasman Sea to SW Pacific (3) | Tasman Sea | 15 | 57 | 1468 |
| | | Tropical Australia & Coral Sea (4) | Coral Sea | 16 | 33 | 10349 |
| | Mid South Tropical Pacific (2) | | -- | 17 | 44 | 2818 |
| | Open Atlantic, Indian, & Pacific oceans (2) | Offshore & NW North Atlantic (4) | -- | 18 | 26 | 7591 |
| | | Offshore Indian Ocean (5) | -- | 19 | 43 | 3486 |
| | | Offshore W Pacific (6) | -- | 20 | 40 | 4678 |
| | | Offshore S Atlantic (6) | -- | 21 | 33 | 5512 |
| | | Offshore mid-E Pacific (7) | -- | 22 | 36 | 1217 |
| | | Tropical E Atlantic (6) | Gulf of Guinea | 23 | 57 | 992 |
| | S South America (2) | Argentina (3) | Rio de La Plata | 24 | 45 | 1651 |
| | | Chile (3) | -- | 25 | 68 | 584 |
| | S Africa, S Australia, & New Zealand (2) | S Australia (6) | South Australia | 26 | 40 | 2158 |
| | | S Africa (5) | -- | 27 | 45 | 6700 |
| | | New Zealand (6) | -- | 28 | 33 | 3126 |
| North West Pacific | | | N W Pacific | 29 | 47 | 2551 |
| Southern Ocean | | | Southern Ocean | 30 | 17 | 4256 |

**Fig. 1** Classification hierarchy and number of species belonging to the mapped biogeographic realms. The classification hierarchy from this analysis is compared to the clustering of seas and oceans as shown on a blue background. The red text in column 1 denotes the realms defined by pelagic-only species and clustered at 1% similarity. The red numbers in parentheses in columns 2 and 3 indicate further similarity index levels, e.g., 3 = 3% similarity between 5° areas in that region. The percentage of unique species indicates how distinct the realm was from the others; cells with yellow background represent > 40%, while cells with peach colored background represent > 50%. Comparisons with previous studies are in Supplementary Table 7. E, east; N, north; S, south; Spp, species; W, west

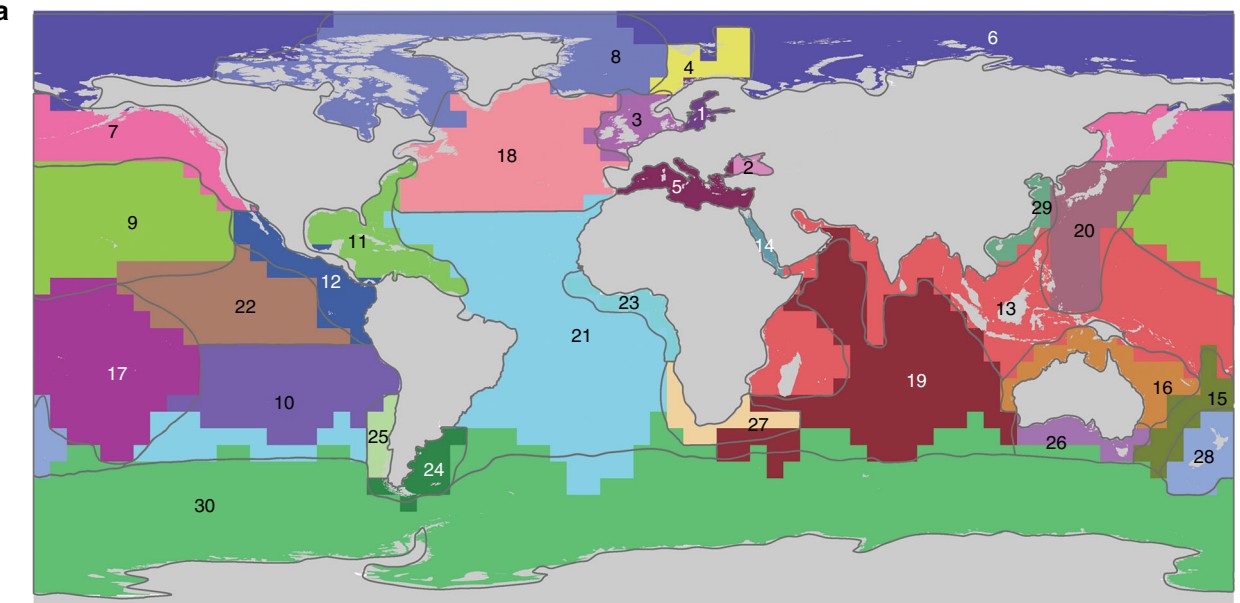

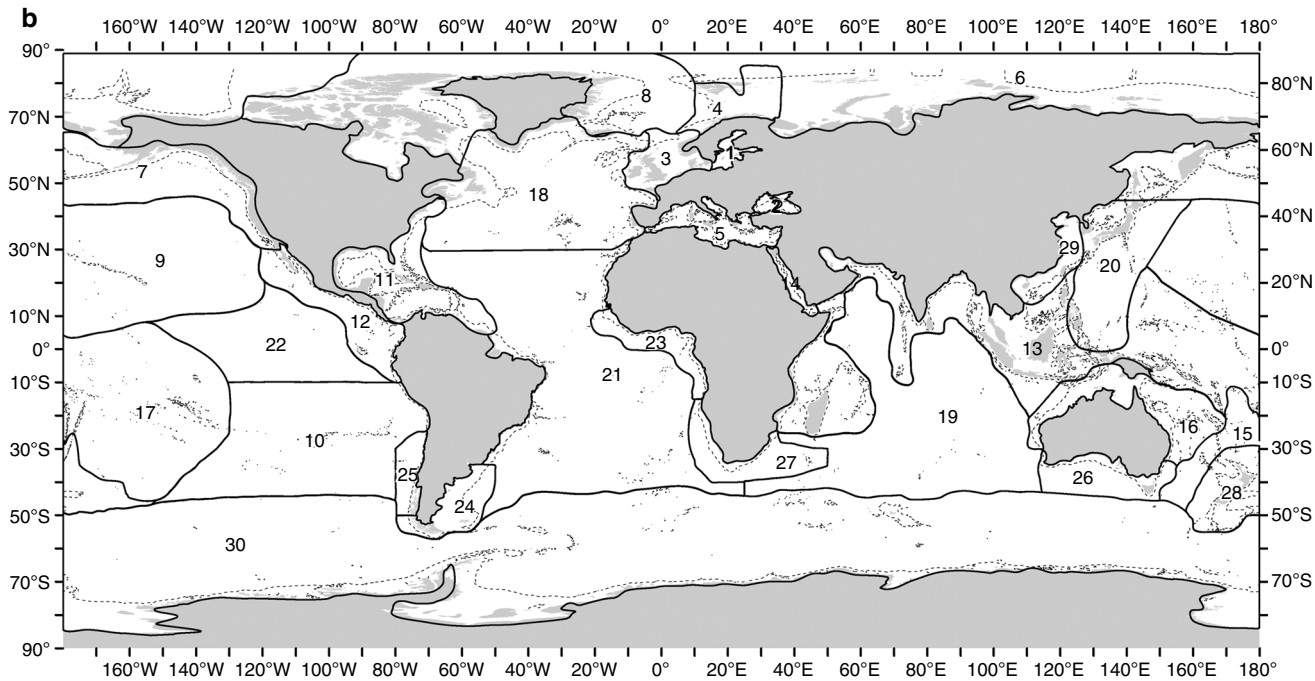

**Fig. 2** The biogeographic realms as numbered 1–30 in Fig. 1. **a** Shows realms (denoted by solid lines) overlaid on the original 5° latitude-longitude cells. Realm boundary smoothing included following the Southern Ocean 10°C annual average sea surface temperature sub-Antarctic Front. **b** Shows the 1000 m depth contour as a dashed line

The African-Asian land bridge separated biogeographic realms at the 1% level, but the Central American land bridge at 3%. The cluster analysis of seas and oceans similarly found that the Tropical West Atlantic (including the Caribbean and Gulf of Mexico) and Tropical East Pacific (including the Gulf of California) were more closely grouped (i.e. related in species composition) with the Indo-West Pacific than with the Atlantic or North Pacific seas, respectively (Supplementary Figs 1 and 2). It also showed that the seas in the outer Mediterranean were a distinct group, but sometimes placed within a larger north-east Atlantic group than the inner Mediterranean seas, suggesting they may form a Lusitanian group as proposed by Ekman[1]. Analyses with alternative indices and cell sizes produced a similar biogeography, but sometimes did not distinguish the Baltic and/

or Black Seas or New Zealand realms (Fig. 3). While the Infomap's bioregion network theory algorithms did not distinguish the Black Sea, they did extend the Caribbean realm down the coast of Brazil to about Rio de Janeiro (Fig. 3).

**Endemicity.** The top 100 most-widespread species in 5° cells were comprised of 27% pelagic megafauna and 72% plankton; and in the seas and oceans were 46% fish and 23% other vertebrates (birds, mammals, and turtles), 14% zooplankton, and 10% phytoplankton (Supplementary Table 2). The most widespread species, the planktonic foraminiferan *Globigerinita glutinata* Egger, 1895, was recorded in 589 (28%) of the 2065 c-square cells (Supplementary Table 3). The proportion of taxa in more than 50

cells that were primarily benthic and pelagic was 3 and 17% respectively, further showing the more widespread distribution of pelagic than benthic taxa. Thus, species-rich benthic taxa such as

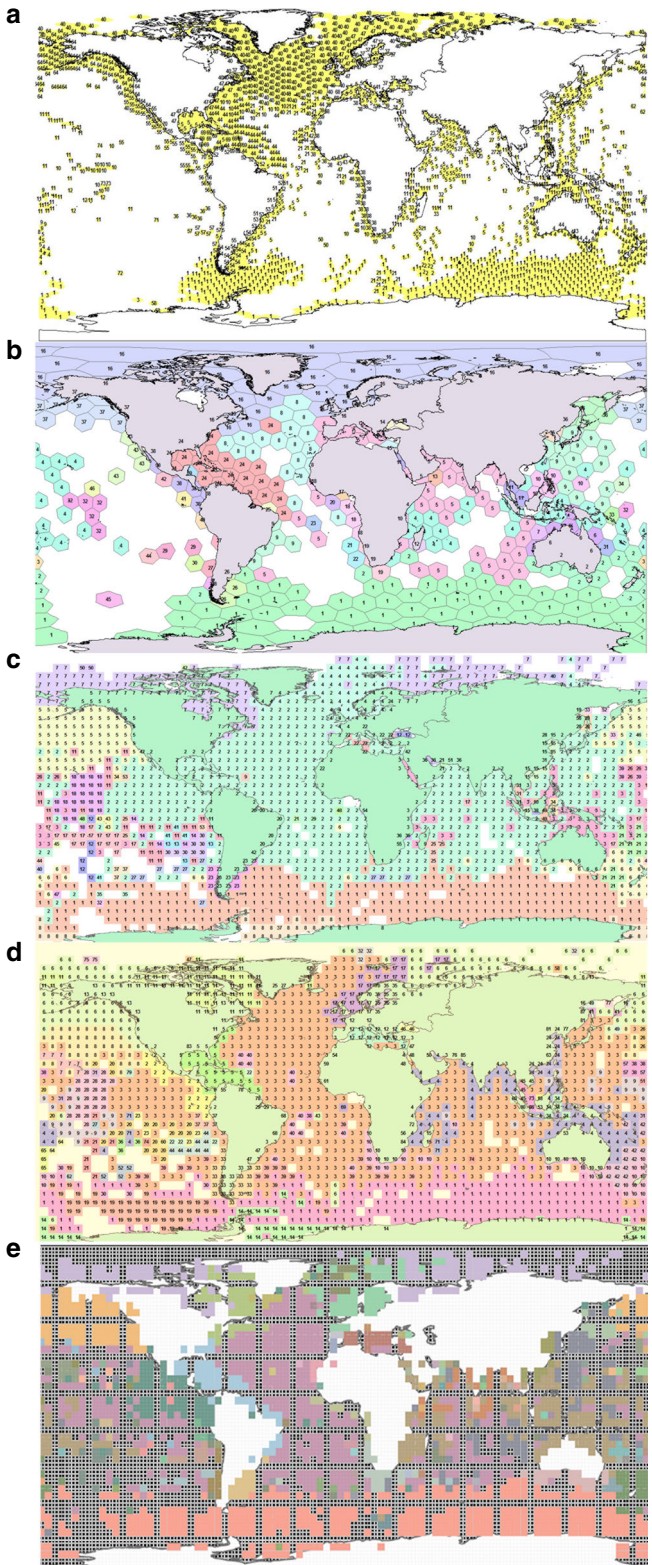

**Fig. 3** Alternative analyses of marine biogeographic realms with alternative dissimilarity metrics and cell sizes. **a–c** Used Beta SIM index and **d** Sorensen's index of dissimilarity. **a**, **b** Used 50,000 km$^2$ and 600,000 km$^2$ hexagons, respectively. **c**, **d** Used 5° latitude-longitude cells. **e** Used Infomaps[50] to construct the realms

arthropods and molluscs contributed most to endemicity (Table 1).

There were from 192 (Black Sea) to 16,508 (tropical Indo-West Pacific) species per realm (Fig. 1). The number of species unique to each realm ranged from 3 to over 4000 (Table 1). Most of these species were arthropods (mostly benthic crustaceans), molluscs, chordates (mostly demersal fish), and cnidarians, followed by annelids (mostly polychaetes) and echinoderms. The species in the Baltic and Black Seas were freshwater and brackish tolerant. In the Mediterranean (15) and New Zealand (28) realms the proportions of nematodes and bryozoans were notably higher than elsewhere. There were on an average $4268 \pm 132$ and $277 \pm 75$ species unique to each of the biogeographic realms and 'seas and oceans', respectively (mean $\pm$ 95% confidence limits). This is conventionally reported as percent endemicity to adjust for species richness. The average percent endemicity was thus, 42% $\pm 5$ (range 17 to 84%) and 11% $\pm 5$ (range 0 to 41%) for the realms and seas, respectively. The realms with the highest percent of unique species were the Black Sea (84%), Red Sea (74%), Chile (68%), Inner Baltic Sea (63%), South-East Pacific (59%), Tropical East Atlantic, and Tasman Sea—New Zealand areas (57%) (i.e., realms numbered 2, 14, 25, 1, 10, 23 & 15), with ten between 40 and 47%, eight between 30 and 36%, and five ≤ 27% (Fig. 1).

## Discussion

We identified 30 biogeographic realms with a minimum of 17% and an average of 42% endemicity, significantly above the threshold of 10% endemicity that has been proposed for a geographic area to qualify as a biogeographic region[11]. Percent endemicity was about four times higher for the biogeographic realms proposed here compared to the 'seas and oceans'; indicating that our realms were the better representation of endemicity.

All measures of endemicity are indices that are sensitive to the data set from which they have been calculated. Further sampling may find endemic species to be more widespread, but it will also find more species, and the number of endemics increases with species richness as found in the present study. Clearly, sampling effort is very unequal between regions of the ocean[24, 26], and tens of thousands of species remain to be discovered in some of the most species-rich areas of the oceans[23]. These yet to be discovered species will generally be more endemic, because widespread species are discovered earlier[27]. Thus, they may subdivide and refine the boundaries of the realms found here rather than change their general location.

Our genus-level analysis produced a similar but less well-defined classification of realms. Reuda et al[6]. conducted genus-level cluster analysis of terrestrial amphibians, birds, and mammals and defined similar biogeographic regions to Holt et al.[5], who used the same data at species level. At least for marine species, what defines a genus is more arbitrary than species and subject to greater change of expert opinion over time. Thus, species may be reclassified under new genera multiple times by different authors. As a consequence, using higher taxa may add error to analyses of biogeographic endemicity.

A small number of unique species can distinguish biogeographic areas even where sampling is incomplete. For example, islands tend to have less species than continents (due to isolation and/or area related effects), but sometimes higher endemicity (due to isolation). Inventories of marine species of Hawaii and New Zealand estimated 11 and 51% endemicity, respectively[28, 29]. Regional assessments of marine biodiversity estimated (without a full species inventory) endemicities of < 10% for the Baltic Sea, Mediterranean, China, and Japan, but 28% for Australia and Southern Africa, and 45% for Antarctica[29]. In the Caribbean, 49%

**Table 1 The percentage of 'endemic' species in the most species-rich phyla that occurred in more than one 5º cell and only one realm**

| Phylum | No. species | Arthropoda | Mollusca | Chordata | Cnidaria | Annelida | Echinodermata | Bryozoa | Rhodophyta | Porifera | Nematoda | Ochrophyta |
|---|---|---|---|---|---|---|---|---|---|---|---|---|
| No. species | 19782 | 4812 | 4432 | 3535 | 1684 | 1358 | 1222 | 651 | 459 | 379 | 310 | 272 |
| *Realm* | | | | | | | | | | | | |
| 1 | 8 | 13 | | **63** | | | | | | | | |
| 2 | 3 | **67** | 33 | | | | | | | | | |
| 3 | 1257 | **22** | 18 | 7 | 5 | 9 | 2 | 5 | 9 | 5 | 7 | 5 |
| 4 | 22 | **50** | 23 | | 9 | 14 | | 5 | | | | |
| 5 | 627 | **23** | 24 | 7 | 5 | 12 | 3 | | 1 | 4 | **21** | |
| 6 | 124 | **40** | 14 | 6 | 5 | 2 | 3 | 13 | | 1 | | 3 |
| 7 | 1973 | **25** | **20** | 16 | 7 | 16 | 6 | 1 | 4 | 2 | | 1 |
| 8 | 29 | **34** | | **34** | | 14 | | | | 7 | 10 | |
| 9 | 274 | 15 | 10 | **31** | **23** | 5 | 12 | | | 1 | | |
| 10 | 105 | **33** | 10 | **39** | 2 | 2 | 10 | | | 2 | | |
| 11 | 4093 | **28** | **23** | 16 | 10 | 10 | 7 | 2 | | 3 | | |
| 12 | 374 | 11 | 6 | **58** | 9 | 2 | 14 | | | | | |
| 13 | 3278 | **22** | 19 | **24** | 12 | 5 | 5 | | 4 | 2 | 1 | 2 |
| 14 | 38 | 5 | 5 | **58** | **26** | | 3 | | | | | |
| 15 | 4 | **25** | | **25** | | | **25** | | | **25** | | |
| 16 | 1214 | 16 | **25** | **39** | 11 | 1 | 2 | 2 | | 1 | | |
| 17 | 121 | **68** | 9 | 15 | 7 | | | | | | | |
| 18 | 281 | **30** | 16 | 12 | 12 | 1 | 1 | 5 | | 2 | 2 | 3 |
| 19 | 19 | **37** | 5 | **21** | 5 | | | | | 11 | | 0 |
| 20 | 315 | **22** | 17 | 9 | **20** | 1 | 19 | | | 1 | | 3 |
| 21 | 422 | **32** | 15 | 14 | 11 | 12 | 3 | | | 1 | | |
| 22 | 19 | **58** | | **26** | | | 11 | | | | | |
| 23 | 67 | 19 | 16 | **51** | 9 | | 3 | | | | | |
| 24 | 240 | **31** | **27** | **28** | 5 | 3 | 2 | 3 | | 1 | | |
| 25 | 19 | **47** | | 5 | **21** | 16 | 5 | | | | | |
| 26 | 192 | 4 | 15 | **54** | 6 | 3 | 1 | 2 | 8 | | | 3 |
| 27 | 1546 | **24** | **46** | 15 | 3 | | | | 7 | | | 2 |
| 28 | 1019 | 12 | **22** | 9 | 5 | 6 | 10 | **31** | | 1 | | |
| 29 | 519 | **25** | **56** | 1 | | 5 | 14 | | | | | |
| 30 | 1580 | **32** | 12 | 8 | 7 | 6 | 13 | 7 | | 2 | 2 | 3 |
| Mean % | | **29** | 16 | **23** | 8 | 5 | 6 | 3 | 1 | 2 | 1 | 1 |

Values > 20% endemicity are emphasized in bold.

endemicity for coastal and 10% for pelagic fishes has been found[30]. Complete species inventories for realms, regions, and countries will enable more accurate calculation of rates of endemicity.

The most widespread (cosmopolitan) species in the present study were pelagic, but of two contrasting groups. The first were planktonic microorganisms that disperse passively without energetic costs, either in water or attached to animals or drifting materials, and can be very abundant in samples. The distributions of many bacterial, protozoan, and microalgal species are more associated with habitat conditions than geography[10]. In contrast, the wide-ranging, but less abundant megafauna (fishes, birds, mammals, and turtles) may swim or fly across oceans. Larger fish tend to have larger geographic ranges[31]. The widespread nature of both pelagic groups will mean they have low endemicity and little influence on the delimitation of the biogeographic realms. Thus, when we analyzed the data using only widespread and pelagic species we found fewer biogeographic realms (Fig. 1). Similarly, a review of pelagic biogeography suggested that there may be only five pelagic biogeographic realms[32]. Endemicity will thus be most influenced by the species-rich benthic macro-invertebrates.

If the abundance of cosmopolitan species is related to environmental conditions, then ecologically-distinct regions are likely to be found, such as those found for 15 species of pelagic fishes[33]. However, these do not align with biogeographic realms (based on

endemicity), because they reflect habitat suitability for the species within their geographic range.

Pelagic and benthic biota tend to be independently sampled, studied, and reported upon, reinforcing impressions that they may be distinct biogeographically. However, there are more benthic species that spend part of their life-cycle in the plankton (meroplankton) than there are holo-plankton, so this division is artificial[34]. Many species in the plankton will return to the seabed during their life, and thus their planktonic and benthic biogeographies will overlap. In both the pelagos and benthos the most widespread species are the microscopic biota (microbial, meio-fauna, and plankton)[35] and mobile megafauna (e.g., birds, mammals). Thus, these very small and very large taxa have less species globally[36]. The taxa that are the most species rich are the benthic macrofauna such as crustaceans and molluscs[23], and they contribute most to endemicity[28], as our study found (Table 1).

The Black Sea and inner Baltic Sea had a biogeographically distinct biota at a global level due to the influence of freshwater species. This illustrates how salinity determines aquatic species distributions at a global level. However, beyond these brackish seas, salinity varies little in the ocean and thus has no further effect on biogeography.

The first two clusters of seas and oceans (Supplementary Figs 1 and 2) separated an Atlantic-Arctic from an Indian-Pacific-Southern Ocean group, with the notable exception of the tropical west Atlantic (Caribbean and Gulf of Mexico), which clustered

with the Indo-Pacific. The closer similarity of the biota between the tropical west Atlantic and tropical east Pacific, than the Mediterranean and Red Sea, reflected the more recent establishment of the Central American compared to the Asia-Africa land barrier. Thus, continental drift has been a primary factor in determining marine endemicity, as it has on land[5, 6].

It may be argued that ocean biogeography should be considered in four dimensions (e.g. latitude, longitude, depth, and time) rather than the two-dimensional approach taken here. However, two dimensions may be as adequate for marine biodiversity mapping as they are for terrestrial because changes in depth (and altitude) coincide with changes in latitude and longitude, and pelagic and deep-sea species are relatively cosmopolitan compared to benthic and coastal. Just as land fauna and flora form distinct communities with altitude, biogeographic boundaries may also occur with depth. The oceans are often divided into bathyal, abyssal, and hadal zones in recognition that the deep-sea fauna varies with depth[13]. However, where the boundaries are reported can vary by thousands of meters, reflecting the lack of a clear concept of how to distinguish the zones, insufficient quantitative data for analysis, and/or that either the boundaries vary geographically or do not exist[2, 13]. Although oceans are three-dimensional (3D) habitats, most zooplankton show diel vertical migration such that there is no evidence of vertically separated zoogeographic regions in the open pelagic oceans[34]. Similarly, habitats on land are 3D, but all taxa directly or indirectly (e.g. attached to vegetation) connect with the soil during their life. In the ocean, the main vertical zonation appears to be between the well-lit euphotic epipelagic zone where algae and herbivores thrive to a depth of 200 m, to the deeper pelagic zones without plants[37], although a distinct mesopelagic (twilight) zone has been distinguished between 200 m and 1000 m[38].

The open-ocean realms found in the present study reflect a combination of widely dispersed pelagic species and deep-sea species. Deep-sea species tend to have wider depth ranges than coastal species, and abyssal species are largely a subset of bathyal species, although there may be exceptions[10, 13]. Geographic ranges are generally larger for pelagic than benthic, and deep-sea than shallow water, species[10, 39]. Thus, as we found, there are likely to be fewer biogeographic realms in pelagic and deep seas than in coastal areas, so mid-ocean biogeographic realms would be expected to be larger than coastal. The effect of coastal species on our realms is evident where the Maldives extend into the middle of the Indian Ocean (realm 13). Other islands may influence other mid-ocean realms (e.g. 9, 10, 20, 22). Over three-quarters (23, 77%) of the realms found here were based on coastal species' biogeography (Fig. 2).

Most realms were coastal and continental shelf. Where the continental shelf was narrow and/or ice covered, such as on the east coast of South America and in Antarctica, respectively, no coastal realm was distinguished. Offshore realms were larger because they have lower endemicity (and betadiversity) in the pelagic and deep-sea environments. The World Register of Deep-Sea Species has taken 500 m as the boundary between coastal (continental shelf) and deep-sea environments[40], and species richness rapidly decreases below 500 m[10]. Below 500 m, the ocean is uniformly cold, dark, and with low productivity and minimal seasonal variation[41]. Thus, the coastal realms should be considered to extend to the 500 m depth contour. The sediment covered seabed has low slope[42], so the deep-sea is a large, but relatively uniform habitat compared with coastal environments. Thus the deep-sea and its associated pelagos is the largest realm on Earth.

Our data qualify, extend, and for the first time, map, the biogeographic realms proposed by Ekman[1] (Supplementary Table 4). Ekman similarly distinguished the following: Baltic (realm 1), Black (2), and Mediterranean (5) Seas; Arctic Ocean (6); northwest American (7); north American boreal (8); West Indian (11); tropical Pacific America (12); Central Pacific Islands (17); tropical West Africa (23); Humboldt Current region (25); Southern Australia (26); Southern Africa (27); New Zealand (28); northeast Asia (29); and Antarctic (30). We subdivided Ekman's European boreal (3, 4), Indo-West Pacific (9, 13, 14, 20), and Tropical and sub-tropical Australia (15, 16) regions. With the exception of the Rio de La Plata realm (24), most of the additional regions proposed here (18, 19, 21, 22) are deep-sea and pelagic; Ekman lacked data for these regions.

Based on a review of current knowledge, Spalding et al.[17] proposed a hierarchical set of coastal Realms, Provinces and Ecoregions, where the first two were considered biogeographic realms in the sense of distinct biota and high endemicity. In contrast, Ecoregions were based on environmental conditions and other factors, and so with two exceptions, we found no evidence for biogeographic differences between them. Our biogeographic realms were a close match to 9 of their 11 Realms, 9 of their 62 Provinces, and 2 of their 232 Ecoregions (Supplementary Table 4). However, the latter two (Baltic and Black Seas) included freshwater species. Consideration of the coastal Realms and Provinces together with the pelagic and deep-sea provinces showed strong similarities with the biogeographic realms found here, reflecting our integration of biogeography's for coastal, deep-sea, pelagic, and benthic environments. Half of our realms were closely related to the proposed pelagic[18] and deep-sea regions[13] (Supplementary Table 4).

Kulbicki et al.[43] distinguished Indo-Pacific, Tropical Eastern Pacific, and Atlantic realms by cluster analysis of 169 checklists of 6316 species of coral reef fishes. These equate to our realms 13, 12, and 11 + 21 + 23. Within the Kulbicki realms were regions that mapped to our realms 9 (Hawaiian central north Pacific), 10 (south-east Pacific), and 17 (mid-south Pacific). Thus, our realms were supported by their analysis and provide some additional biogeography. Where the boundaries of Kulbicki et al. and our realms do not align may reflect the limitations of the data sets that boundaries may differ between taxa and/or that boundaries are wide. Keith et al.[44] compared range maps of 719 species of shallow water scleractinian corals in the Indian and Pacific Oceans to environmental conditions. They distinguished eleven faunal regions which geographically aligned to six of our realms.

Our analysis thus provides empirical support for many of the marine biogeographic realms proposed based on reviews that synthesized taxon-specific and regional knowledge. It further illustrates that one global classification may be realistic and suggests close spatial relationships between the pelagic and deep-sea realms.

The spatial scale of our analysis at 5° latitude-longitude cells may have obscured biogeography within isolated bays or narrow bathyal and hadal depth zones proposed in previous studies. Thus, it is also possible that there will be more realms eventually distinguished than we have found. Abyssal and hadal endemics appear to exist[13], but may not have been sufficiently sampled or represented in the present data to form distinct biogeographic realms. Some regional studies have proposed biogeographic regions within the realms proposed here, such as in southern Africa[45], the Mediterranean[46], Caribbean[30], bathyal and hadal depths[13], eastern North America[47], and many other areas[11]. Analysis of deep-sea hydrothermal vent molluscs in all the world's oceans only found the Mediterranean to be biogeographically distinct from the Indian, west and north Pacific, and Gulf of Mexico regions[48].

Our exploration of alternative similar coefficients and cell sizes (Fig. 3) supported the present results. However, network theory analysis[49, 50] suggested that our Caribbean realm may extend

down the coast of Brazil and merits further analysis with additional data. Cluster analyses of the distribution of 70 species of seagrass and 77 species of razor clams distinguished 11 and 16 regions, respectively, some of which were very small[51, 52]. The relatively low number of species in these studies may explain why they found less than the 23 coastal realms in the present study. In some cases, such as Spalding et al.'s[18] pelagic and Watling et al.'s[13] deep-sea provinces, the regions were primarily based on environmental criteria such as currents, fronts, and gyres for the pelagic and bathymetry, temperature, and particulate organic carbon flux for the deep-sea. The biogeographic boundaries in these and the present study are best considered a hypothesis that should be tested as more species distribution data become available.

The spatial resolution of the present study means that the biogeographic boundaries may be 10° or 1200 km wide. Considering depth, the 500 m depth contour may be a suitable general boundary for coastal to offshore realms[10]. The geographic boundaries were coincident with land barriers to species dispersal, low-salinity seas with freshwater species, and coastal and offshore environments separated by depth. Further research is required to determine what environmental factors explain other realm boundaries. Whether boundaries will change as species change their distributions in response to climate change and human mediated species introductions remains to be seen. For example, species have colonized the Mediterranean from the Red Sea through the Suez Canal[46] and will colonize the Atlantic from the Pacific as the polar ice retreats[53, 54]. With climate change, sea temperature will change mostly in high northern latitudes[55], and so richness is predicted to increase there[56, 52], although species may also change their depth distribution[57, 58]. While changing species distributions will change richness, community composition, and ecosystems, whether they will change the relative location of biogeographic boundaries remains unknown.

While our choice of similarity index (Jaccards) has been the most popular in biogeography analyses[3], indices that are not influenced by species richness, that can distinguish between gradient and nestedness patterns of species turnover (e.g. refs [59, 60]), and alternative methods using network theory[49, 50, 61], have recently been developed. We found our findings robust to alternative similarity indices and spatial units, including using a 2015 version of the data from OBIS (Fig. 3). Similarly, Mouillot et al.[60] found several indices applied to 122 species of Indo-Pacific coral reef fishes separated out a west Indian Ocean region from the Indo-West Pacific. We encourage new analyses using these and related measures in biogeography. Such studies will need to consider the limitations of using primary data vs. species ranges, and computational challenges (e.g. our data matrix was 65,056 species by 2056 locations). Moreover studies that are less than global in taxonomic and geographic scope need to account for possible boundary effects. For example, species apparently endemic may occur outside the study area or habitat. A coral reef fish species may also occur on rocky reefs and coastal rocky reef fish can occur amongst deep-sea cold-water coral reefs[62]. In addition, while species distribution models can predict species ranges (e.g. refs [58, 63]), they may not be appropriate for determining biogeographic realms and boundaries[49]. Our preliminary analysis of modeled species ranges from AquaMaps[64] returned patterns mirroring the environmental variables used to generate the models rather than patterns of endemicity. Another limitation of SDM is that they tend to be applied to widespread species rather than the endemic species that determine biogeography.

The present analysis of 65,000 species across all oceans and higher taxa, both pelagic and benthic, provides the most taxonomically integrated and first map of global marine biogeographic realms based on standardized analysis of primary data (Fig. 1). It complements a similar approach that mapped terrestrial realms[5]. If we add Antarctica to the 11 terrestrial realms and consider that 29% of the planet is land[42], then there is a similar proportion of terrestrial (12 in 29%) to marine (28 in 71%) realms per area. That our findings extend previous studies that used more limited data supports the realms mapped here. The results showed greater species endemicity in coastal than offshore environments, the role of land barriers, depth, and salinity in separating realms, and how 28 fully-marine realms were nested amongst 6 realms based on pelagic species only (Fig. 1). The realms provide a biologically relevant geographic context and hypothesis for understanding the evolution of life on Earth. For more applied studies, they can aid the design of networks of Marine Reserves, monitoring change in biodiversity (including fisheries), predicting the effects of climate change, and are biologically relevant regions on which to report on the state of the world's biodiversity.

## Methods

**Data sources.** The data were obtained from OBIS on 27th July 2009, comprised 815 data sets (Supplementary Tables 5 and 6), 110,000 nominal 'species', and 19.2 million location records, of which 18.1 million had species names. Prior to analysis, the species names were matched to the World Register of Marine Species[65, 66] and were manually inspected for errors. The lack of validation of some names could be because at the time of comparison WoRMS contained about 160,000 of the expected 230,000 described marine species. Inspection of the non-validated names also identified further synonyms and misspellings, as well as entries that were not complete species names, such as a genus name followed by a letter (a, b, c), sp., spp., the names of geographic places, or descriptions of specimens (e.g., unidentified, juveniles, and males). This reduced the species names from 110,000 to 93,000. Initial analyses using the 93,000 names resulted in some seas not being classified with nearby areas although the overall geographic pattern was similar. The use of the 65,000 species (Supplementary Table 7) thus appeared adequate for this global scale analysis and reduced spurious results due to synonyms from analyses. These represented one-third of all described marine species[23]. Because global scale analyses for selected taxa produced poor spatial coverage, and most species were geographically rare (Supplementary Tables 2, 3, 7), we only report results for the entire data set.

We matched the latitude and longitude coordinates for the OBIS species records to the seas after cleaning the data for taxonomy and excluding locations on land. Some data in OBIS were located on land, typically because they were geo-referenced from a place name (e.g., Russia, Australia) without knowing a more precise location. Following preliminary analyses, we selected 5° c-squares as a compromise between the lack of spatial resolution provided by 10° and the computing challenges, and increased number of empty squares in a much larger 1° data set. There were 2056 sample areas of 5° latitude (550 km)—longitude ($\leq$ 550 km) cells using the c-squares geographical indexing system in OBIS[67]. Cells with questionable records were omitted from the analysis. Regardless of how many times a species was recorded in a location, it was represented by one c-squares record. This significantly reduced the amount of occurrence data for analysis. Because the spatial indexing process was automated in a Geographical Information System, this may exclude coastal locations for a species where the geo-referencing was not precise enough. We also excluded five c-squares that preliminary analyses indicated had anomalous data: i.e., latitude-longitude coordinates 0, 0 (the Gulf of Guinea), one c-square in the Alboran Sea (0 longitude), and two in the Gulf of California. Seasonal changes in animal distributions would not affect the biogeographic realms distinguished here, because if species migrated between areas they would be equally recorded for both, i.e., the range of a species encompasses wherever it occurs regardless of season. Thus, the biogeographic realms encompass and integrate the seasonal changes in species' distribution and abundance.

**Data analysis.** To statistically test for geographic structure in the data, we reduced the size of the data set through aggregating records into the international standard seas and oceans map (International Hydrographic Organisation 1953), available from www.marineregions.org, and tested the statistical significance of the similarities between each sea area. Each sea and ocean area was exclusive, so for example, the 'North Atlantic' excluded the seas around it. These data were compared using PRIMER v6[68] because it included tests of statistical significance between sea areas (SIMPROF) and between groups of areas (ANOSIM). The SIMPROF test compared the results of the cluster analysis to the mean which was calculated by randomising the order of the species and re-analyzing the data (Supplementary

Fig. 1). The test then identified which groupings of sea areas were significantly ($P < 0.05$) similar (i.e. not random). A pre-requisite for the ANOSIM test was that adjacent sea areas were first grouped, and then the cluster analysis result was compared with what groupings would arise randomly (by permutation). This found a statistically significant hierarchical relationship between the geographical areas (Supplementary Fig. 1).

Jaccard's coefficient was used to compare the number of species common to a pair of geographic areas in proportion to the total and unique number of species in both areas, i.e. = 100*[(number of species in both areas A and B)]/[(number species in both areas) + (number species unique to area A) + (number species unique to area B)]. This and closely related coefficients are also the most commonly used indices of "species turnover" in biogeography and "beta diversity" in macroecology (e.g. refs [3, 5, 39, 69, 70]). We also explored alternative coefficients (e.g. Sorensen's, Bray-Curtis, Beta SIM, and Infomaps bioregions), including using both the 2009 OBIS data and a similar 2015 data set from OBIS in hexagons used by Chaudhary et al.[26], and found negligible difference in results (Fig. 3). This type of coefficient was necessary for the present study because it is not biased by species absences; i.e., the similarity between location A and B is independent of area C. This is important because the occurrence of species in the source database was strongly influenced by sampling effort, so our analysis excluded absences. The similarity coefficients were clustered using the group-average algorithm rather than single-linkage (nearest neighbor in a cluster) or complete-linkage (furthest neighbor) to also reduce the affect of sampling bias.

The seas were also clustered using the R-programme (http://cran.stat.auckland.ac.nz/), where the significance of the cluster hierarchy was indicated by bootstrapping (re-sampling) the data set 1000 times, and then representing the number of times a pair of areas clustered together as a percentage on the dendrogram (Supplementary Fig. 2). This produced the same groupings of seas as with PRIMER. Multi-dimensional Scaling (MDS) plots on the species occurrence data in the seas and oceans had '2D stress' values near 0.2 indicating difficulty in displaying the data in two dimensions, and so the data were presented as dendrograms.

The c-squares are based on latitude ($5° \approx 550$ km) and longitude ($5° \approx \leq 550$ km) grid, and thus their area decreases away from the equator. However, our analyses were not noticeably affected by the reduction in area of 5° cells towards the poles, as found in previous species similarity studies (e.g. ref. [5]). Analyses using equal area hexagons of two sizes and related similarity indices also found the same geographic clustering (Fig. 3). This is because cluster analysis compares differences between samples and is relatively insensitive to sample size. Thus, it takes few species to show that areas with no species in common are different. For example, marine, terrestrial, and freshwater environments represent 71, 28 and 1% of the planet area, but 15, 77 and 8% of its species, and it would take only a few species to sample to distinguish these were different[9]. Furthermore, all realms encompassed six or more cells except for the Baltic, Black, and Red Seas which had two, three, and two 5° cells, respectively.

The groups of 5° cells distinguished at particular levels of similarity were numbered and visualized on maps of the world (Supplementary Fig. 3). Contiguous areas of the same group were then progressively delimited as biogeographic realms at coefficient levels of 1, 2, 3, 4, 5, and 6 %. Note that this is a similarity index, not the actual percentage of species in common. At higher percent similarities there were few geographically coherent groups of squares. Cells within the same group number that were not adjacent to each other were not used to delimit smaller biogeographic realms. Some groups suggested subregions within the (a) coastal Indian Ocean region and the (b) narrow regions that stretched latitudinally between the North Pacific and tropical Pacific and the Southern Ocean and regions to its north. In contrast, the open-ocean region extended into the temperate and tropical Atlantic, Indian, and Pacific Oceans, and even the south-east Mediterranean Sea. At the 6% level this open-ocean region still covered the South Atlantic, a large area of the mid-east Pacific, included areas in the Mediterranean, and scattered around in the mid-tropical Pacific, bordering the Southern Ocean.

When few species are recorded in a cell, it is likely that they will be common plankton and/or nekton. Because these taxa are relatively cosmopolitan, such cells may arise in unconnected parts of the ocean. Thus, when there were less than four 5° cells surrounded by another group, they were subsumed into that larger group. Vilhena and Antonelli[49] similarly merged clusters with few cells into their adjacent regions. This process produced a set of areas congruent with the groupings of seas and oceans, and thus defined the biogeographic realms proposed here.

The 10% most widespread species in the data set were all pelagic and provided sufficient global cover for cluster analysis. They were clustered to provide a comparison between the biogeography of pelagic-only and all species (Fig. 1). The realms were distinguished by the species unique to each realm, some of which may occur in only one 5° cell. To identify the species that would characterize each realm, we selected species that were only recorded in one realm and occurred in more than one 5° cell (to exclude the rarest species). This resulted in a list of c. 20,000 species (Supplementary Data 1).

**Data availability**. The primary data used here are freely available from OBIS (www.iobis.org). The aggregated species by 5° cell matrix finally used in the data analysis is available from Figshare at https://figshare.com/s/e11b3f7769ef353c6262 and DOI 10.17608/k6.auckland.5086654.

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

## Acknowledgements

We thank: Brian McArdle (University of Auckland) for helpful discussion regarding multivariate analyses and measuring species turnover; Muhammad Sulayman and Syed Husnain Naqvi (University of Auckland) for installing OBIS 2007 database and exporting data in various formats for preliminary data analyses; Tony Rees (CSIRO Tasmania) advice on c-squares and OBIS database structure; Bart Vanhoorne (Flanders Marine Institute) for export of OBIS 2009 species by IHO seas matrix and matching names to the World Register of Marine Species; Yunqing (Phoebe) Zhang, Bill Stafford, and Edward Vanden Berghe (Rutgers, State University of New Jersey) for downloads of OBIS data and metadata; Bob Clarke and Ray Gorley (PRIMER-E Ltd) for advice on the statistical methods; Kristin Kaschner regarding AquaMaps; Irawan Asaad for tips on data management; and Allen Rodrigo (University of Auckland), Mark Gibbons (University of the Western Cape), Jesse Ausubel (Rockefeller University), Pat Halpin (Duke University), Jean-Philippe Lessard (Concordia University), and Ben Holt (Marine Biological Association, Plymouth), for helpful discussion. We further acknowledge that without the efforts of the members of the OBIS community this extraordinary mega-database would not exist. This study was part-funded by a grant from the Alfred P. Sloan Foundation to Memorial University as part of the Census of Marine Life.

## Author contributions

M.J.C.: Conceived, designed, cleaned the data, conducted some analyses, and wrote the paper. P.T., P.S.W., A.K.L.C. and C.C.: Prepared the data and ran analyses, A.K.L.C. and Z.B.: Conducted the mapping.

## Additional information

**Competing interests:** The authors declare no competing financial interests

