## [Peer Review File · Nature Communications]

Reviewers' comments:

Reviewer #1 (Remarks to the Author):

In writing this review, I have followed the explicit questions asked in the Nat. Comm. guide to referees so have generally focused on the bigger picture.

The major claims of the paper are that there are 30 marine biogeographic realms globally. This is the first quantitative analysis to delimit these regions at a global scale across all (found in OBIS) taxa. Previous attempts to delimit such regions are either not at a global scale, not quantitative, or based on a subset of marine taxa. Therefore, the idea of this paper is of general interest to others in the field.

However, I am not sure how strongly this paper will influence thinking in the field because, although it provides a new description, it does not provide new understanding. There are no attempts to explain the pattern in terms of the particular groups of species present in different regions and their attributes, environmental / climatic conditions, or historical biogeographic / evolutionary context. This expansion would greatly strengthen the paper and help determine whether the groupings make biological sense or whether they are more a reflection of the sampling bias present in OBIS, which is notoriously problematic data (although the authors have worked hard to quality check as far as possible).

The relation to previous literature is largely accurate but in some places reads too much like a textbook account of the background to realm delineation and would benefit from instead leaning more towards literature that justifies why it is useful to do this and how it can forward our understanding of global diversity patterns. There are also some errors in the reference to previous work (e.g., publication date and number of provinces identified in Keith et al).

The methods are described in detail in the Supporting Information and I feel some of this would be better included within the main text. I found it hard to follow the logic for some of the explanations so that could also be improved, particularly for the explanation of similarity index choice (e.g., p.35. "This type of coefficient was necessary for the present study because it is not biased by species absences; i.e. the similarity between location A and B is independent of area C "), and some of the language could be made more clear (e.g., the use of "levels" for different meanings within the same sentence). I would also like to know how sensitive the analysis is to choice of similarity metric. Regarding spatial effects, it would also be useful to explicitly test the influence of area on the groupings given, as the authors note, that the area of the 5 degree cells changes from equator to poles.

Returning to the OBIS data, it would be useful to think about how the type of data for different species groups affects its distribution. For instance, some distributions will be based on interpolation across sample data and assume all cells within the range extent are occupied, whereas other data will be point samples. How could this affect the relative distributions and species clustering, particularly if different taxa tend to be different methods of generating distribution data? This also feeds into my other concern – does it make sense to lump all species together from plankton to fish? I'm not convinced by the authors' justification (but perhaps could be if it was expanded) and I think it might mask really interesting differences that could help tease apart the underlying explanations for the patterns.

Thanks

Sally Keith

Reviewer #2 (Remarks to the Author):

The major claims of the manuscript are novel, namely that the work is a "global map of marine biogeographic realms based on statistical analysis" (p. 2). A statistical study on a global marine bioregionalisation is long overdue and deserves to be published.

While this is an achievement comparable to that of Holt et al. (2013), it lacks a historical context in the introduction. For instance, the authors state: "While the occurrence of marine fauna and flora clearly differs between parts of the oceans, whether biogeographic boundaries, and thus definable realms of endemism, exist has not been clear. Consequently, major reviews of marine biogeography did not provide or map boundaries at global scales and stated that there was little evidence for any" (p. 3). This is not true. Marine biogeographic regions have been proposed and mapped since the 19th century, and several after Ekman (1953) have appeared in the late 1960s and 1970s (although Ekman was by far the most relevant). Please change the opening two lines.

I haven't reviewed the statistical section as this is not in my area of expertise.

Malte C. Ebach

Reviewer #3 (Remarks to the Author):

The authors have analyzed the distributions of 65,000 species of marine animals and plants in order to produce a new biogeographic scheme for the world's oceans. The reason given for this project was that previous reviews did not map boundaries at global scales. However, world maps with biogeographic boundaries clearly marked were published by James Dana (1853), Edward Forbes (1856), Samuel Woodward (1856), John Briggs (1974), Briggs (1995), and Briggs and Bowen (2012). The book by Briggs (1974) carried forward the works by Ekman (1935, 1953) and divided the four climatic zones into regions and provinces. Provinces were defined by the possession of at least 10% endemic species. Each successive series of maps and endemism information was improved as new revisions and other systematic works appeared. Rather than building on past research, the authors seek to introduce a new system using an analysis of the OBIS data base. Although such data bases often contain nomenclatorial errors, the authors claim they checked their information with the World Register of Marine Species. But the Register is a work in progress and so far only the North Atlantic can be said to be well known. This suggests that the Register may not be complete to the extent that it can be depended on for the information about all 65,000 species. If this is the case, I suggest that the authors confine their analysis to the North Atlantic. Also, they should examine carefully the results of previous workers and indicate where differences occur.

Response to referees

>> We thank the referees for the time they have put into reading our paper and suggestions.

Reviewer #1 (Remarks to the Author):

In writing this review, I have followed the explicit questions asked in the Nat. Comm. guide to referees so have generally focused on the bigger picture.

The major claims of the paper are that there are 30 marine biogeographic realms globally. This is the first quantitative analysis to delimit these regions at a global scale across all (found in OBIS) taxa. Previous attempts to delimit such regions are either not at a global scale, not quantitative, or based on a subset of marine taxa. Therefore, the idea of this paper is of general interest to others in the field.

However, I am not sure how strongly this paper will influence thinking in the field because, although it provides a new description, it does not provide new understanding. There are no attempts to explain the pattern in terms of the particular groups of species present in different regions and their attributes, environmental / climatic conditions, or historical biogeographic / evolutionary context. This expansion would greatly strengthen the paper and help determine whether the groupings make biological sense or whether they are more a reflection of the sampling bias present in OBIS, which is notoriously problematic data (although the authors have worked hard to quality check as far as possible).

>> We agree it is a good idea to define the species that characterise each realm. This is easier said than done because of the large number of species. Eventually we settled on providing a list of the ~20,000 species unique to each realm in a new Supplementary Material file, and a summary table of their contribution in the main paper.

With regard to how this paper advances understanding, we note that

- (1) We provide the first fully global map of marine species biogeography with the highest reported rates of endemism (42% on average).
- (2) It is the first attempt to use objective data to map realms.
- (3) It is the first integration of all species, rather than selecting for particular taxa or environments.

We show:

- (4) that realms are larger in the open ocean than coastal areas,
- (5) that realms could be nested with five pelagic realms, and
- (6) the most widespread species are pelagic microscopic and megafauna.

Finally the close relationship of the results to previous qualitative (and partial) assessments of coastal and other biogeography's suggests the results are biologically robust. The Supplementary Material also include analyses showing that the data are statistically robust when clustered by seas and oceans. This confirms the strong geographic structure in the data.

The relation to previous literature is largely accurate but in some places reads too much like a textbook account of the background to realm delineation and would benefit from instead leaning more towards literature that justifies why it is useful to do this and how it can forward our understanding of global diversity patterns. There are also some errors in the reference to previous work (e.g., publication date and number of provinces identified in Keith et al).

>> thank you for spotting the citation error. We have corrected it. In paragraph four of the Introduction we mention practical applications of realm classifications in nature conservation and environmental reporting and management and cite examples that provide more details.

The methods are described in detail in the Supporting Information and I feel some of this would be better included within the main text. I found it hard to follow the logic for some of the explanations so that could also be improved, particularly for the explanation of similarity index choice (e.g., p.35. "This type of coefficient was necessary for the present study because it is not biased by species absences; i.e. the similarity between location A and B is independent of area C "), and some of the language could be made more clear (e.g., the use of "levels" for different meanings within the same sentence).

>> Thank you for this feedback. We have moved some of the methods into the main paper as suggested. We have clarified the wording to explain that we did not use absence data. We removed the use of the word 'level' for statistical similarity on page 36 so it is only used for taxonomic levels now.

I would also like to know how sensitive the analysis is to choice of similarity metric.

>> We conducted analyses with related similarity coefficients, such as Bray Curtis and Sorensen's. There is no theoretical reason to choose one over the other and for presence-only data they are almost identical. Thus we choose the simplest coefficient and the one used in almost all other biogeographic clustering studies. We note some new measures that have appeared since we conducted this analysis in the Discussion.

Regarding spatial effects, it would also be useful to explicitly test the influence of area on the groupings given, as the authors note, that the area of the 5 degree cells changes from equator to poles.

>> The 5° sample areas were chosen to reflect the spatial density of available data, smaller cells resulted in many areas with little to no data. The database was also structured based on latitude-longitude. They thus limit how small the realms can be. The Red, Baltic and Black seas comprise 2-3 cells each but these were clearly distinct from others. The NW Pacific, Chilean and Mediterranean had 6-7 cells each and all other realms 9 or more. Five degree cells vary from [150 km² in the Arctic to 3,075 km² at the equator. They are all much smaller than the realms so any combination of sample areas at a similar resolution should produce the same clustering. Sample size varies regardless of the cell size because of varying data availability. Because the clustering method selects cells that have species in common only it is relatively insensitive to sample size compared to parametric analyses. We now cite a similar study on terrestrial biogeography (Holt et al. in *Science*) which also found such clustering methods insensitive to cell area. Were this study to be repeated, it would be now be possible to use equal area hexagons. However they may be criticised in being more numerous near the equator than the poles and thus being insensitive to biogeographic variation towards the poles. However, the strong geographic pattern in the present results and coherence with previous studies shows that the methods have not been biased by cell size.

Comment [MJC1]: Basher do you have areas of each realm for comparison

Returning to the OBIS data, it would be useful to think about how the type of data for different species groups affects its distribution. For instance, some distributions will be based on interpolation across sample data and assume all cells within the range extent are occupied, whereas other data will be point samples. How could this affect the relative distributions and species clustering, particularly if different taxa tend to be different methods of generating distribution data? This also feeds into my other concern – does it make sense to lump all species together from plankton to fish? I'm not convinced by the authors' justification (but perhaps could be if it was expanded) and I think it might mask really interesting differences that could help tease apart the underlying explanations for the patterns.

>> In this case no distributions are based on interpolated data. We only used point observations of species occurrences. We did start looking at selected taxa but this reduced the amount of available data so that global patterns were not clear. We cite taxon specific studies in Discussion. Each taxon may have its own biogeographic pattern based on its evolutionary history. Taxa could also be classified by habitat, environment, and traits. While interesting this would be another paper and the purpose of this paper is to study species richness overall.

Thanks
Sally Keith

Reviewer #2 (Remarks to the Author):

The major claims of the manuscript are novel, namely that the work is a "global map of marine biogeographic realms based on statistical analysis" (p. 2). A statistical study on a global marine bioregionalisation is long overdue and deserves to be published.

>> Thank you.

While this is an achievement comparable to that of Holt et al. (2013), it lacks a historical context in the introduction. For instance, the authors state: "While the occurrence of marine fauna and flora clearly differs between parts of the oceans, whether biogeographic boundaries, and thus definable realms of endemism, exist has not been clear. Consequently, major reviews of marine biogeography did not provide or map boundaries at global scales and stated that there was little evidence for any" (p. 3). This is not true. Marine biogeographic regions have been proposed and mapped since the 19th century, and several after Ekman (1953) have appeared in the late 1960s and 1970s (although Ekman was by the far the most relevant). Please change the opening two lines.

>> We did not mean our text to be interpreted in that way. To clarify this we have revised the sentence to say "major reviews by Ekman (1953) and Briggs (1995) of marine biogeography did not provide or map boundaries at global scales and stated that there was little evidence for any." Neither of these, nor other studies to our knowledge, published a global map of biogeographic realms but both had regional maps and mentioned biogeographic regions in the text. Of course there are numerous other maps of marine regions (see www.marineregions.org for examples).

I haven't reviewed the statistical section as this is not in my area of expertise.

Malte C. Ebach

Reviewer #3 (Remarks to the Author):

The authors have analyzed the distributions of 65,000 species of marine animals and plants in order to produce a new biogeographic scheme for the world's oceans. The reason given for this project was that previous reviews did not map boundaries at global scales. However, world maps with biogeographic boundaries clearly marked were published by James Dana (1853), Edward Forbes (1856), Samuel Woodward (1856), John Briggs (1974), Briggs (1995), and Briggs and Bowen (2012). The book by Briggs (1974) carried forward the works by Ekman (1935, 1953) and divided the four climatic zones into regions and provinces. Provinces were defined by the possession of at least 10% endemic species. Each successive series of maps and endemism information was improved as new revisions and other systematic works appeared. Rather than building on past research, the authors seek to introduce a new system using an analysis of the OBIS data base. Although such data bases often contain nomenclatorial errors, the authors claim they checked their information with the World Register of Marine Species. But the Register is a work in progress and so far only the North Atlantic can be said to be well known. This suggests that the Register may not be complete to the extent that it can be depended on for the information about all 65,000 species. If this is the case, I suggest that the authors confine their analysis to the North Atlantic. Also, they should examine carefully the results of previous workers and indicate where differences occur.

>> We confirm we did check the species names against the World Register of Marine Species (WoRMS). Every species entry was checked using automated text matching tools and manually. The lead author has spent over a decade developing WoRMS. It is a comprehensive inventory of 240,000 marine species and been fully global since 2008 (it was launched in 2007).

>> We agree that a comparison with previous work is important. The referee is correct that we do not provide a detailed comparison against every previous study. But we do build on previous work by comparing our findings to the most recent reviews by Ekman, Briggs, Bowen, Spalding (coastal benthos), Spalding (pelagos) and Watling (deep-sea benthos). This was in the Table S7 of the Supplementary Material and could be moved to the main paper if preferred.

Reviewers' comments:

Reviewer #1 (Remarks to the Author):

Quantitative delineation of marine biogeographic boundaries is needed and this manuscript provides a useful description of those patterns. However, to reiterate my comments on the previous version, it does not explicitly test any theories or hypotheses that attempt to explain how/why these distinct regions would arise and be maintained, which would be necessary to truly forward our understanding. There are some interesting points in the Introduction with relevant ideas that could be tested, and some interesting conclusions – particularly that the most widespread species are microscopic species and megafauna – but these are not developed.

I also still have some methodological concerns. The justification for the similarity metric used (“the simplest coefficient and the one used in almost all other biogeographic clustering studies”) is not sufficient. As the authors note in the Discussion, there are other metrics (e.g., Baselga’s turnover vs nestedness metrics) that could be more suitable for this purpose and I would at least like to have seen greater justification of why these were not tested. Also, I am not convinced that sampling artefacts are unimportant in the patterns we see here. Could it not be that there are fewer, larger realms in the open ocean because this is where we have the least data?

Reviewer #2 (Remarks to the Author):

I still disagree with the comment that:

"major reviews by Ekman (1953) and Briggs (1995) of marine biogeography did not provide or map boundaries at global scales and stated that there was little evidence for any. Neither of these, nor other studies to our knowledge, published a global map of biogeographic realms but both had regional maps and mentioned biogeographic regions in the text. Of course there are numerous other maps of marine regions (see www.marineregions.org for examples) but they are not based on endemism".

Whether former marine bioregionalisations were based on endemism or not is not in question. It is whether there are existing studies of marine biogeographic areas that have been mapped that provide boundaries at global scales.

Some examples include Forbes (1846), Markus (1933), Schilder (1952), Banarescu (1975), but like the Sclateran-Wallacean areas, they were often in dispute.

Scientists have been attempting to map global marine boundaries for centuries. Yours may be the first global quantification, but it is not the first overall.

The authors correctly state this in their abstract:

"To date, there has been no global map of marine biogeographic realms based on statistical analysis of species distributions." But their second line in the Introduction ignores a centuries long tradition of mapping global marine regions. I would strongly suggest that the authors replace the second line:

"Consequently, major reviews by Ekman (1953) and Briggs (1995) of marine biogeography did not provide or map boundaries at global scales and stated that there was little evidence for any"

with something like,

"Consequently, a centuries old tradition of mapping global marine regions has not produced a single robust bioregionalisation based on empirical evidence (Ekman 1953; Briggs 1995)"

This would make your study far more relevant to the history of marine bioregionalisation.

Malte C. Ebach

Reviewer #4 (Remarks to the Author):

Review of "Marine biogeographic realms and species endemism" by Dr Costello and colleagues, which has been submitted for publication in Nature Communications

I was invited at a late stage to comment on this manuscript, which had already been seen by other reviewers and revised accordingly. I have reviewed both the manuscript and associated files as well as the authors' responses to the previous reviews.

In summary, I think this is a very nice study that clearly deserves publication in Nature Communications. The data is extensive, well curated and seems to have been rather well validated. The results are very interesting and (despite not being a specialist in marine biogeography) seem to make a lot of sense empirically. The discussion of the results, and comparison with previous literature, is excellent and goes at reasonable depth in terms of the patterns encountered and their possible biological explanations. The topic of bioregionalisation is also very timely, so I am very happy to see this reaching the marine biodiversity.

The other reviewers are correct that the authors do not provide detailed explanations on how the marine biogeographical regions evolved, but I think that would constitute a future study once the patterns are outlined and presented. For that, a methodology similar to the one recently presented by Ficetola et al (2017) could serve as a point of departure – and the reference should certainly be included here. (It's worth noting though that the paper was not published by the time this manuscript was first submitted)

But before I would recommend final acceptance, I would encourage a number of improvements on some issues that were not picked up during the previous reviews. These are listed below.

1. The main problem of this kind of bioregionalisation analyses is that it easily becomes "story-telling", since it's often hard to test specific hypotheses or compare one set of biogeogeographical regions with another. However, I think the authors could have paid a bit more attention to possible methodological differences, which could help explaining the discrepancies reported between this and other classifications (eg lines 345-347, and 364-367), i.e. not only deriving from differences in the data used. I realize the great deal of effort and time already put into the analyses and text, but I would strongly encourage the authors to at least try one additional method proposed by myself and collaborators (Vilhena & Antonelli 2014; Edler et al. 2017). We now have a very easy-to-use online tool that should allow a quick exploration of the dataset presented, available at <http://bioregions.mapequation.org/> and described in depth in Edler et al. (2017). I'm not trying to self-promote the method, but several of the issues pointed out by the authors and reviewers as potentially problematic are directly tackled in Infomap bioregions, including:

- the use of a species similarity index, in this case the Jaccard, which we showed on theoretical and empirical grounds to comprise potential pitfalls or lack of biogeographic signal (Vilhena and Antonelli 2015)
- the use of 5-degree grids on the entire dataset. Infomap bioregions allows for adaptive resolution, which means that for grids with higher sampling density the resolution is higher
- The issue of species being sporadically found outside their 'normal ranges' (l. 398-400), which can be avoided in Infomap by setting a threshold of occurrences needed to code for presence in a grid
- The computational challenge (cf line 395), since Infomap has the capacity of analysing very large datasets in short time
- The often subjective decision needed in similarity approaches (l. 145), whereas in infomap the objective identification of a 'best number of clusters' is based on information theory metrics
- The difficulties in defining representative species; infomap provides an easy identification of the most common and the most indicative species for each bioregion

The authors may choose to keep their current results as their preferred choice, which would be fine unless there are large differences between the results and the authors would consider that the Infomap bioregions to make more sense biologically. However, I think that such additional analysis (going into the supplementary material) would help convincing readers on the robustness of the biogeographical regions found in relation to the choice of methodology.

For a more formal comparison between the bioregions obtained through different methods, the authors could calculate the goodness-of-fit between different schemes (cf Edler et al., 2017)

2. To further encourage re-analyses and further exploration of the data, will the datasets used be made publicly available? I could not find the source files in the current submission.
3. Another paper that could be interesting to relate to is Kiel (2016), who explores the connectivity and uniqueness of some deep-water organisms. This would fit in eg. line 290
4. The authors use the terms 'realms', and 'realms nested within realms'. I discuss the lack of a common nomenclature in bioregionalisation, as compared to taxonomic classification (Antonelli 2017). My colleagues and I have therefore advocated for the use of a more neutral term, 'bioregion', rather than the long list of similarly sounding terms (see eg Vilhena and Antonelli 2015 and references therein). This is still a matter of taste, but something to consider: how certain can the authors be of having found realms rather than biomes or ecoregions, when none of these have clear definitions?
5. The authors argue strongly for the use of 'all species regardless of their taxonomic classification' (line 64-65). Even if they choose to show their final results based on such data, I think there are practical and theoretical reasons for challenging this view.

In practice, there are huge sampling biases in relation to taxonomic group, with highest sampling and diversity usually correlating with the activity of experts in a taxon. It would be interesting to see, for instance, how bioregions inferred from well-known and well-sampled taxa (eg teleost fishes?) would compared with those inferred from some microbes or another poorly known taxon, a conclusion also reached by the authors later (l. 254-255).

A similar problem derives from the differences in species diversity. For instance, even if we had perfect sampling and carried out a global analysis, invertebrates would drive all results just because of their high diversity, overriding any patterns shown by eg mammals or plants that are much less diverse but actually very important ecologically.

In theory, why should organisms with vastly different biologies (ecological interactions, reproductive mode, dispersal abilities, size, habit...) show the same overarching pattern? On land, it is very clear that eg birds, plants, fungi, and freshwater fishes show highly different spatial patterns of distribution and endemism. In other words, there might not be 'one-size-fits-all' patterns (Antonelli 2017). Despite that claim, the authors do mention differences encountered by analysing subsets of the data, so I think they could simply tone down that claim and acknowledge the assumptions made and their potential implications.

6. As mentioned above, the discussion on the possible causes or correlates of boundaries should link to the new study by Ficetola et al. (2017), possibly outlining which variables would be interesting to include in such model. Some current claims (eg l. 270-271) were not tested and should be toned down or removed.

Other, mostly minor things:

- typo in 'known', l. 99
- l. 121: how were rare species distinguished from poorly sampled ones?
- how were non-native species treated in the analyses?
- l. 195-196: how can you know if the number of species found really reflected the species richness of realms, rather than sampling biases? For such claim I would expect that a rarefaction analyses would be required. This should affect other conclusions made, such as in lines 199-200
- l 224-225 are speculative
- grammar in l. 256 - 'are reflect'
- l. 278 Reuda > Rueda
- l.295 : what do plants comprise here, eg green algae as well? (but I guess now brown and red algae)?

Literature mentioned above (in addition to the ones already cited in the manuscript):

GF Ficetola, F Mazel, W Thuiller - Nature Ecology & Evolution, 2017. Global determinants of zoogeographical boundaries

S Kiel - Proc. R. Soc. B, 2016: A biogeographic network reveals evolutionary links between deep-sea hydrothermal vent and methane seep faunas

A Antonelli - Nature Ecology & Evolution, 2017. Biogeography: Drivers of bioregionalization

D Edler, T Guedes, A Zizka, M Rosvall, A Antonelli - Systematic biology, 2017. Infomap Bioregions: Interactive mapping of biogeographical regions from species distributions

Good luck with the revision!

Alexandre Antonelli
April 21, 2017

Reviewers' comments:

Reviewer #1 (Remarks to the Author):

Quantitative delineation of marine biogeographic boundaries is needed and this manuscript provides a useful description of those patterns. However, to reiterate my comments on the previous version, it does not explicitly test any theories or hypotheses that attempt to explain how/why these distinct regions would arise and be maintained, which would be necessary to truly forward our understanding. There are some interesting points in the Introduction with relevant ideas that could be tested, and some interesting conclusions – particularly that the most widespread species are microscopic species and megafauna – but these are not developed.

RESPONSE

We agree this mapping or realms is a platform (indeed a hypothesis in itself as stated on line 371) for further research to understand the causes of the patterns. Its production has answered questions about whether such realms and boundaries really exist, how pelagic and benthic regions intersect, that the largest and smallest species dominate the pelagos and are the most widespread, and the role of land, salinity and depth in marine biogeography. Such questions could be re-framed as hypotheses. This was a major effort (e.g. 10 years of data collection in OBIS plus two years of analysis) and is a necessary step in such research. We have addressed the points regarding body size in more depth in a review now published in Current Biology (DOI <http://dx.doi.org/10.1016/j.cub.2017.04.060>). Similarly, these are also avenues for further research.

I also still have some methodological concerns. The justification for the similarity metric used (“the simplest coefficient and the one used in almost all other biogeographic clustering studies”) is not sufficient. As the authors note in the Discussion, there are other metrics (e.g., Baselga’s turnover vs nestedness metrics) that could be more suitable for this purpose and I would at least like to have seen greater justification of why these were not tested. Also, I am not convinced that sampling artefacts are unimportant in the patterns we see here. Could it not be that there are fewer, larger realms in the open ocean because this is where we have the least data?

RESPONSE

The referee provides no reason as to why almost all previous biogeographic studies are at fault for using Jaccard’s index or why it is not appropriate. They reiterate our suggestion that it would be “interesting” to explore alternatives but this is not a study on alternative methods. Further support for the robustness of our analysis is that our results are coherent with previous studies, and with alternative geographic analysis (i.e. using geographic seas and oceans), as noted by other referees.

One must appreciate that the analysis is not about how much data or size of sampling areas but how different species composition is geographically. Of course, there are data gaps as we show in previous papers and the SM, and thus we have subsumed such cells within the realms as described.

We nevertheless have involved an additional co-author who was analysing a more recent version of the OBIS database with 51,000 species to re-run the analyses using different measures of similarity, and using both 5° cells and equal area hexagons.

If smaller hexagons are used than 5° then many cells lack sufficient data for analysis and it is difficult to delimit realm boundaries. If larger are used a clearer pattern is observed which is similar to the present analysis but does not distinguish some of the smaller realms (due to cell size). It thus seems that 5° cells were an optimal size for this analysis considering the available data. Further, cell size had no effect on the clustering method as expected based on understanding of the analytical methods and as found in previous studies. We show examples of these additional analyses below.

Below is Sorenen's index using 5 degree cells which produce the same regions as with the present data (Figure S1).

The use of Sorensen's and Beta SIM similarity indices (below) resulted in the same regions as found here regardless of which size cells are used. However, the former did not distinguish the Mediterranean from the N E Atlantic or New Zealand from adjacent regions. This is likely due to the cell sizes because when using 5-degree cells the same pattern was found.

However, the Beta SIM index (below) did distinguish the Mediterranean.

Beta Sim using 5-degree cells (below) showed less discrimination than either Jaccard's and Sorensens. For example, the Baltic, Caribbean, East Pacific and New Zealand regions were not evident.

Using smaller hexagons lacked spatial coverage (below).

Reviewer #2 (Remarks to the Author):

I still disagree with the comment that:

"major reviews by Ekman (1953) and Briggs (1995) of marine biogeography did not provide or map boundaries at global scales and stated that there was little evidence for any. Neither of these, nor other studies to our knowledge, published a global map of biogeographic realms but both had regional maps and mentioned biogeographic regions in the text. Of course there are numerous other maps of marine regions (see www.marineregions.org for examples) but they are not based on endemism".

Whether former marine bioregionalisations were based on endemism or not is not in question. It is whether there are existing studies of marine biogeographic areas that have been mapped that provide boundaries at global scales.

Some examples include Forbes (1846), Markus (1933), Schilder (1952), Banarescu (1975), but like the Sclateran-Wallacean areas, they were often in dispute.

Scientists have been attempting to map global marine boundaries for centuries. Yours may be the first global quantification, but it is not the first overall.

The authors correctly state this in their abstract:

"To date, there has been no global map of marine biogeographic realms based on statistical analysis of species distributions." But their second line in the Introduction ignores a centuries long tradition of mapping global marine regions. I would strongly suggest that the authors replace the second line: "Consequently, major reviews by Ekman (1953) and Briggs (1995) of marine biogeography did not provide or map boundaries at global scales and stated that there was little evidence for any" with something like,

"Consequently, a centuries old tradition of mapping global marine regions has not produced a single robust bioregionalisation based on empirical evidence (Ekman 1953; Briggs 1995)"

This would make your study far more relevant to the history of marine bioregionalisation.

Malte C. Ebach

RESPONSE

We like this reformulation and have replaced the text with a small clarification:

"Consequently, a centuries old tradition of mapping global marine regions has not produced a single robust regionalisation based on empirical species distribution evidence (Ekman 1953; Briggs 1995)"

We avoid word bioregionalisation as it is so loosely used in the literature that some may say they have done so, for example using ocean colour data..

Reviewer #4 (Remarks to the Author):

Review of "Marine biogeographic realms and species endemism" by Dr Costello and colleagues, which has been submitted for publication in Nature Communications

I was invited at a late stage to comment on this manuscript, which had already been seen by other reviewers and revised accordingly. I have reviewed both the manuscript and associated files as well as the authors' responses to the previous reviews.

In summary, I think this is a very nice study that clearly deserves publication in Nature Communications. The data is extensive, well curated and seems to have been rather well validated. The results are very interesting and (despite not being a specialist in marine biogeography) seem to make a lot of sense empirically. The discussion of the results, and comparison with previous literature, is excellent and goes at reasonable depth in terms of the patterns encountered and their

possible biological explanations. The topic of bioregionalisation is also very timely, so I am very happy to see this reaching the marine biodiversity.

The other reviewers are correct that the authors do not provide detailed explanations on how the marine biogeographical regions evolved, but I think that would constitute a future study once the patterns are outlined and presented. For that, a methodology similar to the one recently presented by Ficetola et al (2017) could serve as a point of departure – and the reference should certainly be included here. (It's worth noting though that the paper was not published by the time this manuscript was first submitted)

RESPONSE

We fully agree with the referee and are very pleased at his suggestions and cited the mentioned papers. Indeed we have been wondering how to include the role of tectonic plate history in shaping marine biogeography. The present analysis shows that it is important across central America and Africa-Asia isthmus, but as most species are coastal it may also be significant more widely. We appreciate the recent (2017) papers he has suggested which we now cite.

But before I would recommend final acceptance, I would encourage a number of improvements on some issues that were not picked up during the previous reviews. These are listed below.

1. The main problem of this kind of bioregionalisation analyses is that it easily becomes "story-telling", since it's often hard to test specific hypotheses or compare one set of biogeographical regions with another. However, I think the authors could have paid a bit more attention to possible methodological differences, which could help explaining the discrepancies reported between this and other classifications (eg lines 345-347, and 364-367), i.e. not only deriving from differences in the data used. I realize the great deal of effort and time already put into the analyses and text, but I would strongly encourage the authors to at least try one additional method proposed by myself and collaborators (Vilhena & Antonelli 2014; Edler et al. 2017). We now have a very easy-to-use online tool that should allow a quick exploration of the dataset presented, available at <http://bioregions.mapequation.org/> and described in depth in Edler et al. (2017). I'm not trying to self-promote the method, but several of the issues pointed out by the authors and reviewers as potentially problematic are directly tackled in Infomap bioregions, including:
 - the use of a species similarity index, in this case the Jaccard, which we showed on theoretical and empirical grounds to comprise potential pitfalls or lack of biogeographic signal (Vilhena and Antonelli 2015)
 - the use of 5-degree grids on the entire dataset. Infomap bioregions allows for adaptive resolution, which means that for grids with higher sampling density the resolution is higher
 - The issue of species being sporadically found outside their 'normal ranges' (l. 398-400), which can be avoided in Infomap by setting a threshold of occurrences needed to code for presence in a grid
 - The computational challenge (cf line 395), since Infomap has the capacity of analysing very large datasets in short time
 - The often subjective decision needed in similarity approaches (l. 145), whereas in infomap the objective identification of a 'best number of clusters' is based on information theory metrics
 - The difficulties in defining representative species; infomap provides an easy identification of the most common and the most indicative species for each bioregion

RESPONSE

We agree the new online InfoMap is very promising and agree with its benefits. We looked into this and realised that our GB's of raw data was too large to upload, and it cannot handle the c-square spatial units used here. However, the first step in InfoMap is to spatially aggregate the data, which we had already done. We thus converted our data to the format required; this took several days of

code development and testing. We were very pleased with the output (below) in the sense that of all the measures (maps above) it most closely agrees with our current map. It also distinguishes the Tasman Sea and New Zealand realms. Two differences are it (a) does not distinguish the Baltic Sea from the NE Atlantic, and (b) it extends the Caribbean realm to the north and north-east coast of Brazil.

Following contact with the InfoMap manager, Dr Edler, we understand that modifications to this method are planned and we will follow up with him in the future to explore collaboration in this regard.

We will thus include the above map and hexagon examples in the SM to show readers that we had looked at alternatives but the clustering using Jaccard's at 5-degree resolution appeared optimal in terms of discrimination of realms. However, we will also note that InfoMaps indicates the Caribbean realm extends southwards along the coast of Brazil so that this region merits more detailed analysis in future.

The authors may choose to keep their current results as their preferred choice, which would be fine unless there are large differences between the results and the authors would consider that the Infomap bioregions to make more sense biologically. However, I think that such additional analysis (going into the supplementary material) would help convincing readers on the robustness of the biogeographical regions found in relation to the choice of methodology.

For a more formal comparison between the bioregions obtained through different methods, the authors could calculate the goodness-of-fit between different schemes (cf Edler et al., 2017)

RESPONSE

Ours is the first global mapping of marine biogeography based on data analysis and so will differ from previous studies. As the other schemes were based on management units and/or expert opinion it is not clear what insights any quantitative comparisons would bring. However, we do show (Table S7) that they generally match well with previous schemes.

2. To further encourage re-analyses and further exploration of the data, will the datasets used be made publicly available? I could not find the source files in the current submission.

RESPONSE

Yes. We were not sure how necessary it is to do this because the source data are freely available and regularly updated and cleaned online through iobis.org. We would recommend future users to use more current versions of data from OBIS and GBIF as these resources will become more comprehensive over time. The sources files we used were tens of GB. After filtering and aggregating to 5-degree cells the data file is 500 MB. We have uploaded the aggregated data file to FigShare <https://figshare.com/s/e11b3f7769ef353c6262> and DOI [10.17608/k6.auckland.5086654](https://doi.org/10.17608/k6.auckland.5086654).

3. Another paper that could be interesting to relate to is Kiel (2016), who explores the connectivity and uniqueness of some deep-water organisms. This would fit in e.g. line 290

RESPONSE

We agree, thank you for this good suggestion and have cited Kiel.

4. The authors use the terms 'realms', and 'realms nested within realms'. I discuss the lack of a common nomenclature in bioregionalisation, as compared to taxonomic classification (Antonelli 2017). My colleagues and I have therefore advocated for the use of a more neutral term, 'bioregion', rather than the long list of similarly sounding terms (see eg Vilhena and Antonelli 2015 and references therein). This is still a matter of taste, but something to consider: how certain can the authors be of having found realms rather than biomes or ecoregions, when none of these have clear definitions?

RESPONSE

We agree, nomenclature is far too loosely used in the literature. We thus defined these terms on page 3 of the MS. We follow the original usage of realms being based on endemism. Biomes are based on plant growth forms (we have a new PhD study mapping marine biomes). The word bioregion has been used in the marine environment for expert drawn maps so we'd prefer to avoid it here to make the distinction. Similarly, there is a global marine ecoregion classification (we cite it too) which is based on a mix of expert drawn and other 'regions'.

5. The authors argue strongly for the use of 'all species regardless of their taxonomic classification' (line 64-65). Even if they choose to show their final results based on such data, I think there are practical and theoretical reasons for challenging this view.

In practice, there are huge sampling biases in relation to taxonomic group, with highest sampling and diversity usually correlating with the activity of experts in a taxon. It would be interesting to see, for instance, how bioregions inferred from well-known and well-sampled taxa (eg teleost fishes?) would compare with those inferred from some microbes or another poorly known taxon, a conclusion also reached by the authors later (l. 254-255).

A similar problem derives from the differences in species diversity. For instance, even if we had perfect sampling and carried out a global analysis, invertebrates would drive all results just because of their high diversity, overriding any patterns shown by eg mammals or plants that are much less diverse but actually very important ecologically.

In theory, why should organisms with vastly different biologies (ecological interactions, reproductive mode, dispersal abilities, size, habit...) show the same overarching pattern? On land, it is very clear that eg birds, plants, fungi, and freshwater fishes show highly different spatial patterns of distribution and endemism. In other words, there might not be 'one-size-fits-all' patterns (Antonelli 2017). Despite that claim, the authors do mention differences encountered by analysing subsets of

the data, so I think they could simply tone down that claim and acknowledge the assumptions made and their potential implications.

RESPONSE

We agree taxa have their own biogeographies. Most of the literature is based on taxon-specific (~limited) analyses that then fail to be generalised across other taxa. We have recently shown this for the latitudinal gradient in marine species (Chaudhary et al. 2016 and 2017. TREE). As the referee correctly points out differences between taxa are also subject to large sampling biases (we also showed this in our TREE 2017 paper). However, overall species distinct biogeographic patterns are clear when sampling bias was accounted for. We absolutely agree this is worth further exploration. The present data set had too many spatial gaps for taxon specific data to be included here (we provided these maps with last response to referees to show this). To address this we are conducting taxon specific global biogeography's that collect new data from literature, experts and museum collections. One on razor clams has been published (Saeedi et al. J Biogeog). Others on fish, polychaetes and amphipods are in progress. We have modified lines 64-65 as suggested.

6. As mentioned above, the discussion on the possible causes or correlates of boundaries should link to the new study by Ficetola et al. (2017), possibly outlying which variables would be interesting to include in such model. Some current claims (eg l. 270-271) were not tested and should be toned down or removed.

RESPONSE

We have now cited Ficetola et al.. We did test the differences between the Black and Baltic Seas (re lines 270-271) and checked the species present; both contained freshwater species that led to their distinctiveness.

Other, mostly minor things:

- typo in 'known', l. 99 [corrected]
- l. 121: how were rare species distinguished from poorly sampled ones? [it was not possible to distinguish how poorly a species was sampled]
- how were non-native species treated in the analyses? [We have made a complete checklist of all introduced marine species at the World Register of Introduced Marine Species (WRIMS) <http://www.marinespecies.org/introduced/> Non-native species represent < 0.8% of all marine species. Also, if they are more widespread than in their native range this will decrease rather than increase biogeographic differences.]
- l. 195-196: how can you know if the number of species found really reflected the species richness of realms, rather than sampling biases? For such claim I would expect that a rarefaction analyses would be required. This should affect other conclusions made, such as in lines 199-200. [we agree and deleted those words.]
- l 224-225 are speculative [yes, we think it is reasonable to expect that more data will 'sub-divide and refine' the present classification]
- grammar in l. 256 - 'are reflect' [corrected]
- l. 278 Reuda > Rueda [corrected]
- l.295 : what do plants comprise here, eg green algae as well? (but I guess now brown and red algae)? [yes, all photosynthetic organisms are consider plants here]

RESPONSES in square parentheses above.

Literature mentioned above (in addition to the ones already cited in the manuscript):

GF Ficetola, F Mazel, W Thuiller - Nature Ecology & Evolution, 2017. Global determinants of zoogeographical boundaries

S Kiel - Proc. R. Soc. B, 2016: A biogeographic network reveals evolutionary links between deep-sea hydrothermal vent and methane seep faunas

A Antonelli - Nature Ecology & Evolution, 2017. Biogeography: Drivers of bioregionalization

D Edler, T Guedes, A Zizka, M Rosvall, A Antonelli - Systematic biology, 2017. Infomap Bioregions: Interactive mapping of biogeographical regions from species distributions

Good luck with the revision!

Alexandre Antonelli

April 21, 2017

RESPONSE

We appreciate this referees in depth understanding of the issues in biogeographic data analysis and most interesting papers 'hot of the press' brought to our attention.

REVIEWERS' COMMENTS:

Reviewer #4 (Remarks to the Author):

I thank the authors for having done such a careful revision of their manuscript in light of my and the other reviewers' comments. I think these revisions helped to further improve the paper. In particular, the new series of analyses show that the results presented are largely robust to the choice of methodology, which is often critical in bioregionalisation. I have therefore no major comments nor corrections, but provide only a few minor additional suggestions:

- If the authors agree, I'd suggest that some of the figures in the appendix (such as S1, S3 and S5) be moved into the main text instead. Not only would they more clearly illustrate the robustness of the results, but also add some colour to the paper (sorry for the silly remark, but true). Given that the journal is online only, I wouldn't expect this to be an issue, and it would make those results more accessible to a broad readership (few people read supporting materials).

- InfoMaps should be 'Infomap bioregions' or just 'infomap'

- Alexander > Alexandre (and thanks for the acknowledgement by the way)

I very much look forward to seeing this paper out!

Best wishes

Alexandre Antonelli

REVIEWERS' COMMENTS:

Reviewer #4 (Remarks to the Author):

I thank the authors for having done such a careful revision of their manuscript in light of my and the other reviewers' comments. I think these revisions helped to further improve the paper. In particular, the new series of analyses show that the results presented are largely robust to the choice of methodology, which is often critical in bioregionalisation. I have therefore no major comments nor corrections, but provide only a few minor additional suggestions:

Thank you

- If the authors agree, I'd suggest that some of the figures in the appendix (such as S1, S3 and S5) be moved into the main text instead. Not only would they more clearly illustrate the robustness of the results, but also add some colour to the paper (sorry for the silly remark, but true). Given that the journal is online only, I wouldn't expect this to be an issue, and it would make those results more accessible to a broad readership (few people read supporting materials).

We have moved two figures to the main MS and another table as suggested.

- InfoMaps should be 'Infomap bioregions' or just 'infomap'
- Alexander > Alexandre (and thanks for the acknowledgement by the way)

Corrected
Journal does not allow referees to be named in acknowledgments

I very much look forward to seeing this paper out! Best wishes
Alexandre Antonelli